# Development of a General Health Score Based on 12 Objective Metabolic and Lifestyle Items: The Lifestyle and Well-Being Index

**DOI:** 10.3390/healthcare10061088

**Published:** 2022-06-11

**Authors:** Octavio Pano, Carmen Sayón-Orea, María Soledad Hershey, Maira Bes-Rastrollo, Miguel A. Martínez-González, J. Alfredo Martínez

**Affiliations:** 1Department of Preventive Medicine and Public Health, School of Medicine, Universidad de Navarra, 31008 Pamplona, Spain; opano@alumni.unav.es (O.P.); mhershey@hsph.harvard.edu (M.S.H.); mbes@unav.es (M.B.-R.); mamartinez@unav.es (M.A.M.-G.); 2Navarra Institute for Health Research, IdiSNA, 31008 Pamplona, Spain; 3Navarra Public Health Institute, 31003 Navarra, Spain; 4Área de Fisiopatología de la Obesidad y la Nutrición, Centro de Investigación Biomédica en Red (CIBEROBN), 28049 Madrid, Spain; 5Department of Nutrition, Harvard T.H. Chan School of Public Health, Boston, MA 02115, USA; 6Department of Food Sciences and Physiology, University of Navarra, 31008 Pamplona, Spain; jalfmtz@unav.es; 7Precision Nutrition and Cardiometabolic Health Program, IMDEA Food Institute, 28049 Madrid, Spain

**Keywords:** well-being, health related quality of life, precision medicine, lifestyle, metabolism, Index

## Abstract

Healthy and unhealthy lifestyles are tightly linked to general health and well-being. However, measurements of well-being have failed to include elements of health and easy to interpret information for patients seeking to improve lifestyles. Therefore, this study aimed to create an index for the assessment of general health and well-being along with two cut-off points: the lifestyle and well-being index (LWB-I). This was a cross-sectional analysis of 15,168 individuals. Internally valid multivariate linear models were constructed using key lifestyle features predicting a modified Short Form 36 questionnaire (SF-36) and used to score the LWB-I. Categorization of the LWB-I was based on self-perceived health (SPH) and analyzed using receiver operating characteristic curve analysis. Optimal cut-points identified individuals with poor and excellent SPH. Lifestyle and well-being were adequately accounted for using 12 lifestyle items. SPH groups had increasingly healthier lifestyle features and LWB-I scores; optimal cut-point for poor SPH were scores below 80 points (AUC: 0.80 (0.79, 0.82); sensitivity 75.7%, specificity 72.3%)) and above 86 points for excellent SPH (AUC: 0.67 (0.66, 0.69); sensitivity 61.4%, specificity 63.3%). Lifestyle and well-being were quantitatively scored based on their associations with a general health measure in order to create the LWB-I along with two cut points.

## 1. Introduction

Over the past few decades, the role of the environment for populational health has gained greater interest among the scientific community as we now know these play a key role for defining daily habits, which lead to non-communicable diseases (NCD) [1]. Physical inactivity alone is attributed with a total of 832 thousand yearly deaths worldwide for its role in cardiovascular and other diseases; and smoking habit is attributed with 23.6% and 14.5% of all forms of cancer for men and women, respectively [1]. Although less studied, environments also influence the way an individual experiences health and disease and, thus, the level of physical, mental and social satisfaction derived from a particular health state, also known as health-related quality of life (HRQoL) [2]. This judicious assessment of health is also determined by factors such as socioeconomic status and early life experiences, but also personal goals and values [3,4]. As such, the role of environments in defining daily habits and perceived health will remain relevant in the coming decades as we continue reducing the impact of NCDs on populational health.

It is clear how daily habits influence biological pathways for health, some of which are related to the immune system or inflammatory markers and promote healthy cellular activity; however, the impact of health perception on health is less studied and yet highly relevant [5]. Clear examples of its importance for health and longevity include: health and well-being have been linked to self-harm attitudes in the adolescence [6]; detrimental lifestyles such as poor sleep [7,8] and unhealthy eating habits [9] negatively impact HRQoL through various underlying mechanisms; and, compared to participants with no impairment, those reporting deteriorated self-perceived health (SPH) saw a RR for mortality of 1.99 (95% CI: 1.64 to 2.42) [10]. Some researchers stress that in some instances, these effects are of significant magnitude and thus cannot be omitted from future research [8,11]. However, clear limitations to the concepts of HRQoL and SPH exist. For one, the associations between health states and HRQoL are not always straightforward; for example, two individuals experiencing similar health conditions could report significantly different HRQoL [12]. This phenomenon, however, has been linked back to the relevance of environments, cultural values and individual experiences to define health, but mainly to acquired lifestyles [4]. Secondly, the assessment of HRQoL is traditionally based on questionnaires that lack clear, interpretable results to guide clinical practice and improve patients’ well-being [13]. For these reasons, and despite the fact that these tools are highly useful for researchers, they are less practical in a clinical setting and among the general population [12].

Traditionally, HRQoL assessment tools focus on the impact of health and disease on either physical or mental well-being domains [14,15]. At this point, a distinction must be made between SPH and HRQoL, the latter is a broader concept that encompasses how an individual perceives their well-being from a physical, mental and social standpoint [2]. Contrastingly, SPH relates to the impact of health on daily activities, social relations and mental well-being. As such and based on the type of questions presented in HRQoL assessment tools, these more accurately reflect SPH [2]. The Short Form 36 (SF-36) questionnaire, the most widely used general assessment of HRQoL, evaluates a total of eight health domains pertaining to either physical or mental component scales including: role limitations due to physical or mental issues, bodily pain, vitality, general health, mental health, social function and physical function [13,14]. In contrast, disease specific assessments include additional health domains, which assess the detrimental effects of treatment, or the level of dependence caused by a disease. However, these tools suffer from similar limitations as general assessment tools, with low interpretability and highly dependent on the individuals’ perception.

Current risk assessment scores focus on measuring biomarker levels and early metabolic changes in order to predict disease incidence or mortality [16]. However, this approach is now being questioned as we lack information on the factors that lead individuals and populations to experience satisfactory and fulfilling lives. This ‘positive epidemiology’ approach centers on HRQoL and well-being and has led to novel insights for health and yet we require further study of the lifestyles and characteristics of individuals living satisfactory lives [11]. In this line, tools for the assessment of life satisfaction and well-being include the Flourishing index and the Ryff questionnaire, which has been associated with a moderate reduction in mortality in a recent review [17,18]. Additional approaches to characterizing states of well-being have been reported, not only for longevity and health, but for particular diseases and risk factors. The Mini-Nutritional Assessment questionnaire, a tool that combines elements of well-being and objective measures such as anthropometric measurements, is a tool for malnutrition screening in elder populations [19]. This simple questionnaire, for use in clinical scenarios, accelerates the assessment of these individuals at their arrival, reducing the risk of in-hospital complications due to malnutrition [19]. Similar tools for the general population might provide additional information on traditional and non-traditional determinants of health, as well as an insight into the ways environments and daily habits promote states of well-being associated with positive outcomes.

One particular example of a lifestyle index has seen great success as a global assessment of health by combining a variety of laboratory and well-being items. The MEDLIFE index evaluates daily habits based on a traditional Mediterranean lifestyle [20]. By including elements such as conviviality during meals and performing group activities, among traditional elements of diet and cholesterol profiling markers, this index is a clear example of the utility of mixed measurements of health and well-being. This lifestyle index has seen strong associations with incidence of metabolic syndrome (compared with unhealthy lifestyles, participants with high adherence to the MEDLIFE index had an OR of 0.29, 95% CI: 0.10 to 0.90) and depression (compared to the lowest quartile of this index, those in the third quartile saw HR of 0.74, 95% CI: 0.61 to 0.89), both of which have been linked to lifestyles and general states of mental well-being [20,21,22]. Missing from this index, however, are other relevant domains of HRQoL such as bodily pain, psychological well-being and mental health, all of which are critical for individuals to experience fulfilling and successful lives [17,23,24]. Further development of similar indices might provide additional insight into the associations between environments and their impact on health perception, but also the way these influence daily habits, which in conjunction could provide more adequate estimators of traditional epidemiological outcomes and determinants of satisfactory lives and lifestyles. Under this premise, it can be hypothesized that individuals experiencing diverse states of health and well-being may reflect distinct habits and particular features, which put them at higher risk of adverse outcomes.

Therefore, our aim was to design an index capable of objectively evaluating health and well-being based on pondered associations between key nutritional, metabolic and lifestyle features with HRQoL. Moreover, two optimal cut-off points were defined for the index according SPH, which classified subjects into three states of lifestyle and well-being. An interactive version of the tool will accompany this work for the reader to estimate their lifestyle and well-being based on the associations found in this sample.

## 2. Materials and Methods

### 2.1. Study Type and Population

This is a cross-sectional analysis nested in the ongoing, permanently open, prospective, dynamic, multipurpose cohort, the “Seguimiento Universidad de Navarra” (SUN) project; ClinicalTrials.gov registry identifier: NCT02669602. Participants include Spanish university graduates who were invited to participate once they had concluded their undergraduate studies; this was considered the baseline for each participant. This cohort sought to establish associations between dietary and lifestyle habits with chronic diseases since December 1999 [25]. Data were collected through self-administered standardized questionnaires at standard interval follow-ups of two years.

As of December 2019, the SUN-Cohort study included a total of 22,894 participants. Cross-sectional analyses were conducted for the 4th year of follow-up; however, sociodemographic and lifestyle characteristics were taken from the baseline questionnaire as shown in Figure 1. As such, the following exclusion criteria were applied: 4 years and 9 months of follow-up were required as shorter follow-ups precluded the completion of the SF-36 questionnaire; SF-36 information was insufficient; failure to answer item 1, item 28, or item 31 of the SF-36 questionnaire; pre-defined energy intake limits (female intake <500 kcal/day or >3500 kcal/day; and male intake <800 kcal/day or >4000 kcal/day); and missing information on other variables of interest such as dietary and lifestyle habits.

### 2.2. Primary Outcome: Health Related Quality of Life (SF-36)

HRQoL was assessed at the fourth year of follow-up using a validated, Spanish version of the SF-36 questionnaire and scored from the RAND version of this questionnaire [9]. The SF-36 uses 36 Likert-scale questions, which are scored on a scale from zero to 100 (100 corresponds to the best possible HRQoL state) to assess 8 health domains. Finally, a global SF-36 score was obtained as the mean score of each of the health domains. Further scoring details and additional measures are described elsewhere [26].

For our analysis, the SF-36 was modified to exclude items 1, 28 and 31 for later use as predictors in the LWB-I. In accordance to scoring instructions, missing items were imputed only if these did not represent more than 50% of the items conforming a health domain and dropped if missing items exceeded this threshold; overall, less than 2% of data were imputed. Item 1 (In general, would you say your health is) of the SF-36 was used as a single-item assessment of SPH and to define cut-off points for our index. Item 28 (During the past 4 weeks, have you felt downhearted and blue?) and item 31 (During the past 4 weeks, did you feel tired?) were items corresponding to the emotional well-being and vitality health domains of the SF-36, respectively.

### 2.3. Sociodemographic, Dietary and Lifestyle Items Included in the LWB-Index

The variables used to create the LWB-I were self-reported data taken from the baseline questionnaire as described in Figure 1. Item selection was based on the literature and on previously described associations between sociodemographic, anthropometric and lifestyle features with HRQoL in this sample [9,22,27]. For the predictor data, categorizations were systematically analyzed in order to identify subgroups at distinct risk of increased or decreased SPH, as described by other reports on the development of multivariate indices [16]. These transformations are described below.

Level of adiposity was calculated using self-reported weight and height, which were then used to calculate body mass index (BMI) using the standard formula and reported in kg/m^2^. In order to demonstrate the reliability of self-reported weight and height data, a validation study was conducted in 2005, in which it was concluded that the data were of sufficient reliability [28]. Smoking status was included as a categorical variable identifying current, former and non-smokers; missing data were imputed and accounted for <1% of the sample. Family history of diseases (FHD) identified the presence of the following diseases in either parent: obesity, hypertension, diabetes, cardiovascular disease (CVD; myocardial infarction or cardiac sudden death) and various forms of cancer (lung, colon, rectum, melanoma and breast). Pre-existing diseases were identified with a single variable for prevalent cases of diabetes, hypertension and hypercholesterolemia; reported as a quantitative discontinuous variable ranging from 0–3. Each of these diseases were initially self-reported by participants, following the diagnosis by a medical practitioner and confirmed through additional questionnaires and official medical records [25].

Dietary characteristics were obtained through a validated, 136-item, self-reported, semiquantitative food frequency questionnaire (FFQ) [25]. From the FFQ, the average consumption of fruit and vegetable (FV) was calculated and reported as the number of servings per day (serv/day). The consumption of sugar products was assessed using the combined consumption of sugar packets, jam and carbonated beverages and categorized as: non-consumers, less than one overall serv/d and those who consumed more than one overall serv/day.

Additional lifestyle characteristics included the assessment of physical activity and insomnia as an indicator of sleep quality [29]. Insomnia was self-reported and categorized as: never experienced insomnia; rarely presenting insomnia; and currently experiencing insomnia or past symptoms of insomnia. The assessment of physical activity mimicked the methods of two notorious cohorts, the Nurses’ Health Study and the Health Professionals’ Follow-Up study, methods that were further validated in this cohort [30]. In short, subjects self-reported the frequency with which they performed 17 activities over the past year, for which a standard metabolic equivalent of task (MET) was known; also known as leisure time physical activity (LTPA). Data were then transformed to METs/h per week and categorized for analysis according to the World Health Organization (WHO) recommendations: where less than 2.5 h/wk of moderate intensity activities (<150 min/wk of activities such as brisk walking or low intensity cycling per-week) was considered as below recommendations; between 2.5 and 5 h/week of moderate/vigorous intensity activities (<300 min/wk of moderate and vigorous activities) was within recommendations; and over 5 h/wk of vigorous physical activity was considered to exceed current recommendations of LTPA [31].

### 2.4. Statistical Analyses

The samples were characterized according to SPH, measured through item 1 of the SF-36, which consisting of five categories that were transformed as follows: *poor* and *fair* were pooled into a “poor” SPH group, *good* and *very good* conformed the “transition” SPH group, and a third category for *excellent* SPH remained independent from the rest. Additionally, two distinct dichotomous variables were created to uniquely identify subjects with poor SPH, and a second for those with excellent SPH; variables that were later used to develop the cut points for the LWB-I based on receiver operated characteristics curves. Baseline comparisons were made across groups of poor, transition and excellent SPH groups using analysis of variance (ANOVA) for normally distributed, continuous data (ascertained through graphical means) and reported as means (SD). χ^2^ distribution was used to analyze categorical variables, which were reported as percentages.

The selection of sociodemographic and lifestyle variables used to create the LWB-I were systematically analyzed, as (1) univariate associations with the modified SF-36 scores and (2) in a multivariate model created through a stepwise method using the following criteria for including a variable (univariate *p* value of <0.20) and for the permanence of said variable in the model (*p* value had to remain <0.30 in the fitted model). Interactions were explored, but ultimately excluded (data not shown) as they did not significantly improve the performance of the model. Once the final set of variables were defined, the *Leave-one-out cross validation* (LOO-CV) method was used to create individual β-estimates for each of the participants in the sample. According to the LOO-CV method, each participant is iteratively removed from the sample and a multivariate linear regression model is fitted with the 12 predictors. The resulting β-estimates are then used to ponder the characteristics of the participant which had been excluded from the dataset and define their LWB-I score. Therefore, this validation method creates a training set comprised by *n* = (15,176 − 1) individuals and a validation set encompassing the entire sample. As the variance in β-estimates were low across all items (data not shown), the tool and tables presented use the estimates of the standard linear regression model. Additional cross-validations methods were conducted, such as the K-folds method (using 5 and 10 validation sets) along with bootstrap β-estimations for the items of the predictive model; no significant differences between methods were found. Reference categories were either the absence or wellness category of a trait (BMI, smoking status, pre-existing diseases, FHD, insomnia and items 28 and 31 of the SF-36) or the category “male” when analyzing sex. Assumptions needed to conduct multivariate model were tested by testing the normality of the residuals as well as their variance homogeneity.

To define categorize the LWB-I, optimal cut points sought to identify participants with poor and excellent SPH. The use of SPH as outcome of interest, is supported by previous reports and in the absence of a gold standard for well-being [32]. ROC and AUC analyses were calculated for the LWB-I as the independent variable and poor SPH or excellent SPH as outcome variables. The following methods were used to identify optimal cut points: (a) Youden index-an optimization of sensitivity and specificity for every value of the LWB-I and (b) The Jacobson–Traux (JT) formula—a method considering the data distribution of LWB-I scores between groups; this last method is recommended in the absence of a gold-standard and psychometric outcomes [33,34]. The JT formula is the following:*c* = ((*m*1 × *s*2) + (*m*2 × *s*1))/*s*1 + *s*2(1)

The formula was used two-fold to identify a lower and an upper cut-point (*c*) capable of distinguishing between the scores of subjects in the poor SPH or the excellent SPH groups. Mean and standard deviation (SD) (*m# s#*) of LWB-I scores were obtained for each of the groups of poor SPH (yes = *m*1, *s*1; or no = *m*2, *s*2) or excellent SPH (yes = *m*1, *s*1; or no = *m*2, *s*2). The sensitivity and specificity were calculated for the newly defined cut-offs and additional exploratory scores (cut-offs that were rounded to the nearest integer). Finally, an interactive tool was created based on the questions used in the SUN study to emulate the assessment of the LWB-I. This application is available as Appendix A. Sub-group analyses were conducted for groups that were under-represented in our sample and to identify significant deviations of the index scores, these included individuals with overweight and obesity and those over 50 years of age. For each subgroup, new LWB-I estimates were obtained and tested using the previously defined cut-points in order to calculate sensitivity and sensibility parameters. Additionally, degree of agreement was calculated between sub-group and whole sample LWB-I estimates using a pondered Kappa index (K-index) analysis.

## 3. Results

After applying the exclusion criteria, the final sub-sample included a total of 15,168 participants (Figure 2). When comparing the excellent and poor categories of SPH (Table 1), individuals with excellent perceived health were younger (34.4 yrs. (SD: 10.9) versus 44.3 yrs. (13.2)), had lower BMI (22.7 kg/m^2^ (3.0) versus 24.7 kg/m^2^ (4.2)), exerted more physical activity (27.4 METs-h/wk (27.7)) compared to those with poor SPH (18.5 METs-h/wk (20.7)) and were healthier overall. Interestingly, in the poor SPH group, 7.3% of individuals consumed no added sugars compared to the 4.5% in the excellent SPH group.

### 3.1. Development of the LWB-I

The multivariate linear regression model (adjusted R-squared of 0.48, *p* < 0.001) consisted of 12 lifestyle habits and features, presented in Table 2, which significantly contributed to the logistic models and the R-squared value of the linear model; data are presented in Appendix A. Direct associations scored positively in the LWB-I and were found for healthy habits such as FV consumption (β = 0.07 (CI 95%: 0.01, 0.12); *p* = 0.017) and exerting above the recommendations of physical activity (β = 0.05 (CI 95%: −0.47, 0.57); *p* = 0.843). In contrast, unhealthy lifestyles were inversely scored by the LWB-I, these included participants that were current smokers (β = −0.86 (CI 95%: −1.24, −0.48); *p* < 0.001) and those with current or past experiences of insomnia (β = −3.46 (CI 95%: −3.90, −3.03); *p* < 0.001).

To demonstrate the scoring process of the LWB-I, we developed the interactive tool found in Appendix A, which uses the estimates from Table 2 to ponder lifestyle and individual features.

From sociodemographic data, female participants scored lower in the LWB-I; β coefficient: −1.13 (CI 95%: −1.47, −0.78). Inverse associations were also found for individuals with pre-existence of three diseases (β = −4.11 (CI 95%: −6.20, −2.02); *p* < 0.001) or those whom both parents presented a FHD (β = −0.45 (CI 95%: −0.86, 0.03); *p* = 0.035). The consumption of ≥1 serv/day of added sugars were inversely scored: β = −1.45 (−2.53, −0.38); *p* = 0.008, as well as participants that did not comply with physical activity recommendations with a coefficient of β = −0.58 (−0.90, −0.27). Finally, items 28 and 31 of the SF-36 had statistically significant coefficients across all categories. Individuals in the category ‘All of the time’ had an associated reduction of −20.45 (−23.53, −17.37) for item 28 and −25.37 (−27.46, −23.28) for item 31.

### 3.2. Defining LWB-I Cut Points

ROC curve analysis, Table 3 and Figure 3, revealed an AUC of 0.80 (0.79, 0.82) when identifying individuals in the poor SPH category and 0.67 (0.66, 0.69) when identifying individuals with excellent SPH group. Sensitivity and specificity for the cut-off points (Table 3) are described for the Youden method, the JT formula and for additional exploratory cut-points. Optimal cut-points were the exploratory lower cut-off 80 points with a sensibility of 75.7% and specificity of 72.3% and the exploratory upper cut-off of 86 points with a sensitivity of 61.4% and specificity of 63.3%. A description of the sample using exploratory cut-offs (lower bound 80 and upper bound 86 points) can be found in Appendix A. The LWB-Index tool is presented as Appendix A using data from the entire sample.

Subgroup analysis, Appendix A, for individuals over 50 years resulted in estimations that did not significantly differ from the estimations of the whole sample. The weighted Kappa-index for the subgroup of age over 50 years was 0.88 (0.87, 0.89) and an agreement of 91.8% with the whole-sample estimates. Similarly, the predictions for the subgroup of overweight and obese individuals were similar to the whole sample estimations of the LWB-I; weighted Kappa of 0.91 (0.90, 0.91) with a 93.8% of agreement with the original estimates.

## 4. Discussion

In this study, an operational definition of lifestyle and well-being was methodologically possible through 12 sociodemographic, dietary and lifestyle items, which quantitatively characterize general health and perceived well-being defined by a modified SF-36 questionnaire. Based on sound statistical methods, β-coefficients served to ponder self-determined lifestyle characteristics in order to calculate the LWB-I. Cut-off points were systematically estimated to distinguish individuals with *poor* (<80 points), *transition* (between 80 and 86 points) or *excellent* (>86 points) SPH and used to describe their particular nutritional and lifestyle characteristics. ROC and AUC analyses revealed that these cut-off points adequately classified the sample in three groups that were distinctively related to detrimental habits in the case of the poor SPH group, and healthful habits for those who reported excellent SPH. Additionally, an interactive tool that calculates the LWB-I accompanies this report and exemplifies the calculation of the index based on the data from this sample (Appendix A).

### 4.1. Lifestyle and Well-Being

Lifestyle and nutritional factors have been loosely studied in relation to subjective determinants of health and health perception [35,36]. Based on the available evidence and results from this study, lifestyle and well-being can be operatively defined using 12 key dietary and lifestyle features within the SUN study. In order to create the single measure of LWB-I, each characteristic was pondered using the β-estimates of the multivariate models for a quantitative assessment of general health and well-being. Developing a classification for our index required the use of SPH in the absence of a gold standard to classify lifestyle and well-being. These methods rely on the premise that individuals with distinct SPH should also differ in their lifestyle and well-being scores; a conceptual framework that has been described by the developers of the JT formula [34,37]. In our sample, initial stratification by SPH and the cut-points that were selected for lifestyle and well-being (the exploratory cut-points of 80 and 86 points) confirmed these clinically significant differences for most of 12 items of the index. Previous attempts at defining cut-points for perceived health are surprisingly compliant with our results; however, our report stands out for its use of the use of a single HRQoL questionnaire and the use of lifestyles features as determinants of general health and SPH [38]. A possible explanation for these trends involves underlying homeostatic mechanisms that regulate well-being [38,39]; however, we can neither prove or disprove this hypothesis, nor was this the goal of our report.

### 4.2. Lifestyle Characteristics in Association with Well-Being

Lifestyle and dietary-pattern assessments tools have been qualitatively associated with HRQoL [9]; however, our index uses a scoring algorithm that ponders these complex associations. The relationship between SPH and lifestyles has been previously demonstrated using a causal inference framework. In the work by Bauldry, S. and collaborators, it was determined that aside from socioeconomic status, adolescent lifestyles and anthropometric features were determinant of SPH and HRQoL in early adulthood [4]. Indeed, our assessment did not study these associations longitudinally; however, it can be inferred that items relating to BMI, LTPA and nutrition such as the single item assessment of fruit and vegetable consumption were valid means of assessing well-being as previous reports have stated [40,41]. Regarding the assessment of diet quality, multi-item or single item assessments have both been described in the past [33]. Irrespectively of the form of measurement, the underlying mechanisms in terms of HRQoL relate to the impact of high-quality diets on well-being. More specifically, these associations could initially stem from the personal satisfaction derived from improving ones eating patterns and in the long-from the innate properties of healthy diets. Similar chains of events have been described in clinical trials that include behavioral counseling for dietary interventions [40,42]. In contrast, sugary products were included for their role as determinants of body composition and glucose metabolism, as well as their negative effects on health and well-being [43,44,45,46]. We observed relatively low consumption of these products across the categories of SPH; hence, only the group of poor SPH was significantly associated with sugary product consumption. We would expect for inverse associations to be consistent in future studies, particularly in samples with higher consumption of similar food items, and thereof, marked detrimental effects on lifestyle and well-being [46,47]. Physical activity on the other hand, is known for its positive impact on health in a dose-response relationship [48,49]. Nevertheless, we found that only the lack of compliance to physical activity requirements had inverse associations with health, suggesting that these benefits are obtained irrespective of the type and duration of the activity. Interestingly, the items for insomnia, tiredness and mood were major contributors to our models. Insomnia in particular could reflect the effect of external stressors that manifest as sleep disturbances as described before [43]. The contributions of items 28 and 31 (mood and tiredness, respectively) on the other hand, have been attributed to changes in vitality and determinants of perceived health [45]. Both of these items served to broadly classify lifestyle and well-being, whereas lifestyle characteristics provide a more precise estimation and help contextualize these subjective components.

### 4.3. Metabolic Characteristics and Their Associations with Well-Being

Previous reports highlight the subtle changes in perceived health prior to the onset of metabolic diseases similar to the changes in metabolic syndrome [40,50]. Thus, BMI, hypertension and hypercholesterolemia were included, and contributed significantly to the model. BMI and HRQoL have bi-directional associations with HRQoL [51]; however, the effects of health on body composition are expected to be limited due to the low prevalence of obesity in our sample [25]. The integration of these comorbidities into a single item may be underestimating the associations for diabetes; however, these data were used in the absence of reliable indicators of glucose homeostasis [52]. Finally, the impairment of well-being scores attributed to recent diagnoses of diseases such as diabetes have been previously described and known to recede over time [43]. These changes could not be explored in this cross-sectional analysis; however, we would expect minor changes of our estimations for individuals with poor health given the low prevalence of diabetes as the only major disease considered in the index [25].

### 4.4. Strengths and Limitations

Overall, the main strength of this index was an intuitive score and interpretation of precluded lifestyle characteristics as potential determinants of low SPH and general health. The 12 items that conform this index can be obtained from most medical records or research datasets, and thus could serve for initial screening of participants and patients in clinical and research scenarios. Based on these items, it was also possible to create a working definition of healthy lifestyles and well-being, a definition that is open for scrutiny and further research as novel studies contribute to this report. The field of application is open to clinical practices as general assessments of health, which allow for the identification of detrimental lifestyles based on the scores obtained for each of the items in the LWB-I. As such, recommendations can be tailored to an individuals’ detrimental lifestyle traits, a feature that had been absent from HRQoL and other SPH indices. A major strength was the successful definition of cut-off points to identify three SPH groups. These categories identified the subgroup of our sample who consider their health to be “poor”, which also corresponded to detrimental daily habits and overall characteristics.

The methods supporting the SUN project have been widely criticized, but also consistently endorsed by the scientific community on their validity. Self-reported outcomes and exposures have been validated and, thus, the results from this study are of sufficient credibility. Furthermore, the sampling of the SUN project was an intended restriction of the sample. This method of restriction allows for the control of socioeconomic status and education degree within the sample parting from the premise that a variable that does not vary within a sample cannot influence other estimates [53]. Indeed, this sampling restriction precludes the SUN project from estimating true prevalence of diseases and impaired SPH; however, it does not prevent us from reaching conclusions regarding biological plausibility and degree of association. Examples of this method are also present in the Nurses’ Health Study and the Health Professionals Follow-up Study, on which this cohort based [54,55]. A major concern was the validity of our results for overweight and obese individuals and the elderly, as these groups were under-represented in our sample. However, subgroup analysis revealed that the estimations and classification of these groups did not significantly depart from our initial results. Regarding the method of evaluating HRQoL, the 2-year difference between well-being and lifestyle measurements could have slightly underestimated our coefficient for age and preexisting diseases. However, our estimations are reliable given the low prevalence of diseases and homogenous socioeconomic status of the sample. The reliability of self-reported data was thoroughly addressed with validation studies.

This index could serve as periodical lifestyle assessments in various clinical scenarios after its validation and to identify early changes in nutritional status prior to the onset of metabolic diseases. Iterative assessments could reveal progressive, clinically significant changes in SPH as lifestyles and habits are modified with potential implications for health over time. We encourage other research groups to analyze these associations in their datasets to contrast with our findings. Going forward, these associations and other characteristics should be explored for a better understanding of lifestyle and well-being and health.

## 5. Conclusions

In conclusion, a set of 12 items describing major sociodemographic, anthropometric, dietary and lifestyle characteristics adequately estimated health and well-being in a sample of 15,168 Spanish individuals. Based on these associations, a definition of lifestyle and well-being as well as an index, the LWB-I, were developed by examining an individual’s features to assess general health. In addition, two optimally defined cut-off points at 80 points and 86 points of the LWB-I were established to categorize samples and populations in order to facilitate the development of preventive measures for subjects with hindering lifestyles and deteriorated well-being. This approach to well-being from the perspective of lifestyles offers a novel insight into the associations between lifestyle and perceived health for patients, health professionals and policymakers alike who seek to improve individual or populational health.

## Figures and Tables

**Figure 1 healthcare-10-01088-f001:**
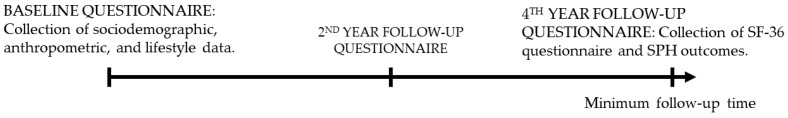
Data origin for the present analysis.

**Figure 2 healthcare-10-01088-f002:**
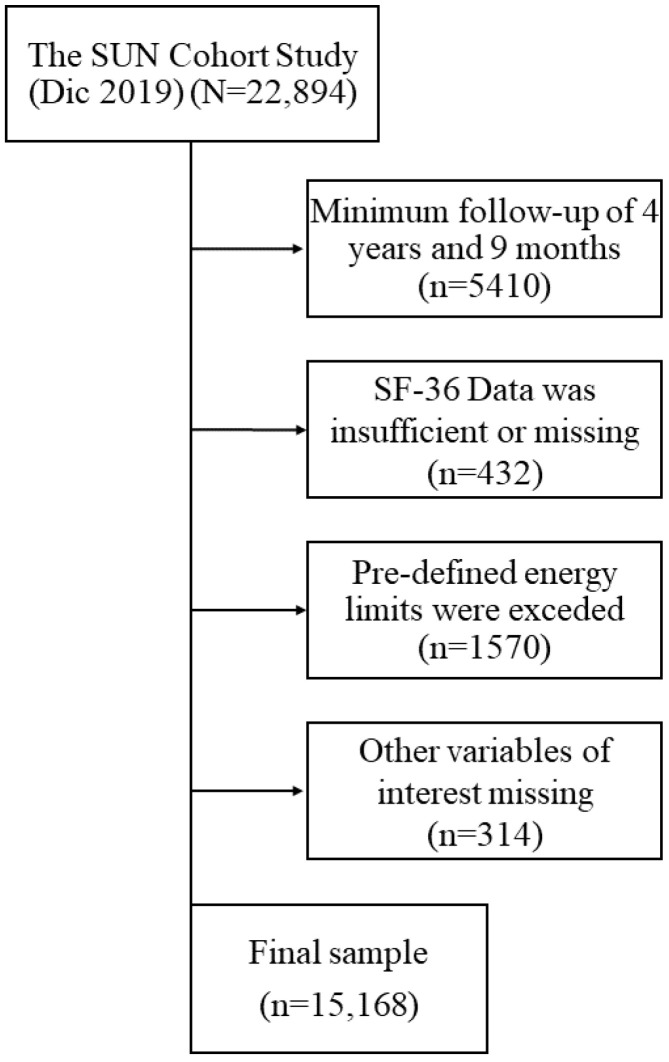
Final sample and exclusion criteria.

**Figure 3 healthcare-10-01088-f003:**
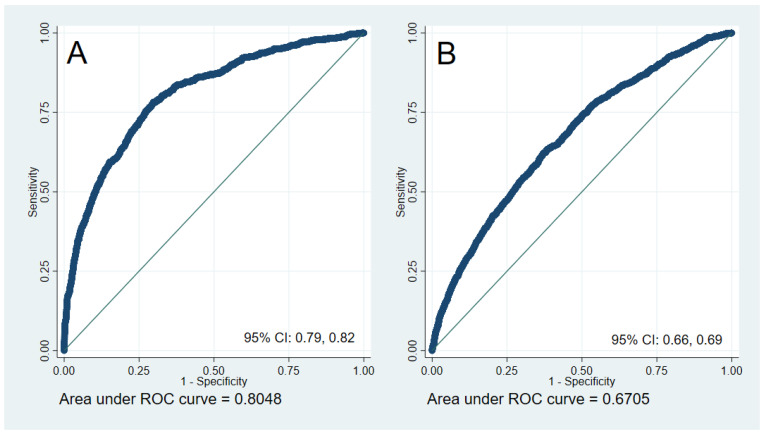
ROC Curves for the Lifestyle and Well-being index using Item Nº 1 of the SF-36 as the outcome. (**A**): Categories Poor and Fair as outcomes (**B**): Category Excellent as outcome.

**Table 1 healthcare-10-01088-t001:** Baseline characteristics of participants categorized by Item 1 of the SF-36 (In general, would you say your health is).

Characteristics	Total (*n* = 15,168)	Poor SPH (*n* = 803)	Transition SPH (*n* = 12,919)	Excellent SPH (*n* = 1446)	*p*-Value
Female Sex (%)	59.8	58.9	59.8	60.0	0.805
Age (years)	38.4 (12.0)	44.3 (13.2)	38.5 (11.9)	34.4 (10.9)	<0.001
BMI (kg/m^2^)	23.5 (3.5)	24.7 (4.2)	23.6 (3.5)	22.7 (3.0)	<0.001
Underweight (<18.5 kg/m^2^; %)	3.7	3.7	3.5	5.4	<0.001
Normal weight (18.5–24.9 kg/m^2^; %)	66.3	55.0	66.1	74.1	
Overweight (25.0–29.9 kg/m^2^; %)	25.4	31.3	25.8	18.4	
Obesity (>30.0 kg/m^2^; %)	4.6	10.0	4.6	2.1	
Smoking status (%)					<0.001
Never	49.3	38.9	48.9	58.8	
Current	21.2	23.0	21.5	18.0	
Former	29.5	38.1	29.6	23.2	
Family history of diseases ^a^ (%)					<0.001
0	36.0	26.5	35.7	44.0	
1	43.7	42.7	44.0	41.4	
2	20.3	30.8	20.3	14.7	
Pre-existing diseases ^b^ (%)					<0.001
0	75.0	59.0	74.7	86.7	
1	19.9	25.3	20.4	12.4	
2	4.6	13.3	4.5	0.8	
3	0.5	2.4	0.4	0.1	
Insomnia (%)					<0.001
Never	34.1	15.2	33.6	48.8	
Rarely	47.4	46.3	48.4	39.2	
Yes	18.5	38.5	18.0	12.0	
Physical activity (METs-h/week)	21.7 (22.7)	18.5 (20.7)	21.3 (22.0)	27.4 (27.7)	<0.001
Fruits + vegetables (serv/day)	4.7 (2.8)	4.7 (3.0)	4.7 (2.8)	4.7 (2.7)	0.789
Added sugars ^c^ (total servings)					<0.001
None	5.4	7.3	5.4	4.5	
<1/day	91.8	87.7	91.9	93.3	
>1/day	2.8	5.0	2.7	2.1	
SF-36 score	82.1 (12.6)	56.7 (16.8)	82.7 (10.7)	90.9 (6.9)	<0.001
SF-33 score	82.1 (12.7)	56.8 (19.9)	82.7 (10.8)	90.7 (6.9)	<0.001

Data is presented as unadjusted means (SD), or percentages for categorical data. Units of measurement are presented along with each variable. Categorization was done according to question 1 of the SF-36. Answers include *poor, fair, good, very good and excellent.* Categories poor/fair and good/very good were pooled according to the researchers’ criteria and group size. *p* values were obtained using χ^2^ distribution for categorical variables and one-way ANOVAs for continuous variables. Prior assessment of data distribution of continuous variables was analyzed using tests for normality and graphical means. Abbreviations BMI: Body mass index; SF-36: Short Form 36 Questionnaire, SF-33: Modified version of the SF-36 excluding items 1, 10 and 28. a: identifies the existence of chronic diseases in both parents ranging from absent (0), present in one parent (1) and present in both parents (2). A detailed list of the included diseases can be found in the main text. b: identifies the number of diseases present for each subject. Diseases include diabetes, hypertension and hypercholesterolemia. c: Pooled analysis of standard servings of sodas including products labeled as “low calorie” (200 cc), sugar (10 g) and marmalade (10 g) were included.

**Table 2 healthcare-10-01088-t002:** Multivariate linear regression models using total SF-33 scores (range from 0 to 100) as the outcome. Description of β-Coefficients (difference in SF-33 for each unit of the independent variable). All variables used to develop the Lifestyle and Well-being Index are included in The SUN cohort.

LWB-I Items	Beta Coefficient (β)	Lower Bound	Upper Bound	*p*-Value
Sex (female vs. male)	−1.13	−1.47	−0.78	<0.001
Age (for each year)	−0.04	−0.05	−0.02	<0.001
BMI category ^a^				
Underweight (<18.5 kg/m^2^)	0.05	−0.74	0.83	0.911
Normal (18.5–24.9 kg/m^2^)	0 (Ref.)	0 (Ref.)	0 (Ref.)	0 (Ref.)
Overweight (25.0–29.9 kg/m^2^)	−0.64	−1.01	−0.27	0.002
Obesity (>30.0 kg/m^2^)	−2.03	−2.75	−1.30	<0.001
Smoking status				
Never	0 (Ref.)	0 (Ref.)	0 (Ref.)	0 (Ref.)
Current	−0.86	−1.24	−0.48	<0.001
Former	−0.56	−0.91	−0.20	0.002
Family History of Disease ^b^				
0	0 (Ref.)	0 (Ref.)	0 (Ref.)	0 (Ref.)
1	−0.14	−0.47	0.19	0.420
2	−0.45	−0.86	−0.03	0.035
Number of Pre-existing Diseases				
0	0 (Ref.)	0 (Ref.)	0 (Ref.)	0 (Ref.)
1	−0.68	−1.07	−0.30	0.001
2	−2.39	−3.12	−1.65	<0.001
3	−4.11	−6.20	−2.02	<0.001
Insomnia				
Never	0 (Ref.)	0 (Ref.)	0 (Ref.)	0 (Ref.)
Rarely	−1.48	−1.81	−1.15	<0.001
Yes, currently, or in the past	−3.46	−3.90	−3.03	<0.001
Physical Activity				
Below recommendations ^c^	−0.58	−0.90	−0.27	<0.001
Recommended ^c^	0 (Ref.)	0 (Ref.)	0 (Ref.)	0 (Ref.)
Above recommendations ^c^	0.05	−0.47	0.57	0.843
Fruits + vegetables (for each serv./d)	0.07	0.01	0.12	0.017
Sugary products ^d^ (serv./d)				
None	0 (Ref.)	0 (Ref.)	0 (Ref.)	0 (Ref.)
Less than 1	−0.32	−0.97	0.34	0.343
More than 1	−1.45	−2.53	−0.38	0.008
Have you felt downhearted and blue? (Item 28, SF-36)				
All of the time	−20.45	−23.53	−17.37	<0.001
Most of the time	−26.22	−28.10	−24.35	<0.001
A good bit of time	−21.07	−21.94	−20.20	<0.001
Some of the time	−10.85	−11.3	−10.38	<0.001
A little of the time	−3.53	−3.87	−3.20	<0.001
None of the time	0 (Ref.)	0 (Ref.)	0 (Ref.)	0 (Ref.)
Did you feel tired? (Item 31, SF-36)				
All of the time	−25.37	−27.46	−23.28	<0.001
Most of the time	−23.39	−24.52	−22.25	<0.001
A good bit of time	−13.83	−14.58	−13.09	<0.001
Some of the time	−7.65	−8.32	−6.99	<0.001
A little of the time	−3.36	−4.02	−2.71	<0.001
None of the time	0 (Ref.)	0 (Ref.)	0 (Ref.)	0 (Ref.)
Constant	98.11	96.97	99.24	<0.001

The coefficients used to develop the index were obtained through multivariate linear regressions using the “*Leave one out*” method. The modified version of the SF-36 (SF-33) was used as a dependent (predicted) variable, whereas the variables described in this table were included as independent variables in a single model. For categorical variables, reference categories were set to the absence of the condition or defined by literature-based recommendations. Each β-Coefficient represents the pondered association between each variable or variable category and an individual’s health and well-being. a: For comparison purposes, the proportion and number of cases in each category are as follows: underweight 3.7% (548 cases); normal weight 66.3% (9920 cases); overweight 25.4% (3807 cases); and obesity 4.6% (649 cases); b: The item designates the number of parents that present any of the diseases described in the main text; c: Below recommendations—less than 2.5 h/wk of moderate intensity activities, recommended-2.5 and 5 h/wk of moderate/vigorous intensity activities; d: Pooled analysis of standard servings of sodas, including artificially sweetened beverages (200 cc), sugar (10 g) and marmalade (10 g) were included.; Above recommendations-over 5 h/wk of vigorous physical activity.

**Table 3 healthcare-10-01088-t003:** ROC and area under the curve (AUC) analysis of the Lifestyle and Well-being Index and Sensitivity and Specificity of the Cut-off points.

**Lower Cut-off**			
AUC (CI: 95%) ^a^	0.80 (0.79, 0.82)		
**Method**	**Proposed Cut-off**	**Sensitivity (%)**	**Specificity (%)**
Jacobson-Traux formula	77.9	69.0	77.3
Youden index	80.7	77.8	70.3
Exploratory *	80.0	75.7	72.3
Exploratory _b_	81.0	78.6	69.2
**Upper Cut-off**			
AUC (CI: 95%) ^a^	0.67 (0.66, 0.69)		
**Method**	**Proposed Cut-off**	**Sensitivity (%)**	**Specificity (%)**
Jacobson-Traux formula	84.3	71.0	52.9
Youden index	86.0	61.6	63.1
Exploratory *	85.0	66.1	56.9
Exploratory _b_	86.0	61.4	63.3

Logistic regression models were used to analyze the ability of the index to identify the outcome of interest. a: For the lower cut-off, the outcome was poor self-perceived health (SPH), defined by item 1 of the SF-36 (categories: Poor and Fair versus Good, Very Good and Excellent). *: Excellent SPH was set as the outcome for the upper cut-off, defined by item 1 of the SF-36 (categories Excellent vs. Poor, Fair, Good and Very Good). b: Exploratory cut-offs were obtained by upwards/downwards rounding of values obtained by the previous methods. The cut-off points were defined using Youden’s index, a standardized formula developed by Jacobson and Traux in 1991 based on mean (SD) HRQoL scores [32]. The estimated cut-off points were determined to the nearest thousandth decimal.

## Data Availability

Data will be made available under petition and only after the approval by the chair of the Department of Preventive Medicine and Public health of the University of Navarra and other members of The Seguimiento Universidad de Navarra Cohort Study.

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
