# Peer review of "Development of a General Health Score Based on 12 Objective Metabolic and Lifestyle Items: The Lifestyle and Well-Being Index"

_healthcare, 2022, doi:10.3390/healthcare10061088_

Round 1
Reviewer 1 Report
1. The abstract should be rewritten in a way that summarize the purpose methods results and conclusion in a simple to understand form. The abstract in its present form is very confusing.
2. The introduction should include more examples of previous similar attempts and make the case why there is a need for new methods of well being assessment and what this methods should add to previous methods.
3. The methods section is very lengthy and hard to follow. Also there are some components which should be presented in results section not in methods section. Like the number of excluded patients from the study because of different reasons. You have repeated this information in the results section.
3. More than a third of your sample has been excluded mostly because of lack of specific data. This is serious and shows low quality in data gathering which can be a source of bios. This should be mentioned as a shortcoming of the present study.
4. You have used a special sample of only university graduates. This make it hard to extend your results to the whole population. This is a serious sampling limitation of your study which should be mentioned as a shortcoming.
5. Many sentences in results section should be moved to methods section. Results section should only give the results not clarifying how each results have been calculated. This clarifications should be moved to methods section.
6. In discussion section tell the reader what is the advantage of your methods related to previous methods. Is it simpler to perform? Is it more sensitive?
6. The English text is very hard to understand and follow in its present format. The entire manuscript need to be edited regarding both grammar and clarity.
Author Response
Reviewer 1
- The abstract should be rewritten in a way that summarize the purpose methods results and conclusion in a simple to understand form. The abstract in its present form is very confusing.
We appreciate the reviewers time and commentary. We agree that our abstract could be improved and simplified. As such, it has been entirely reworded in order to facilitate its reading by the audience of the journal. Page 1 Line 19-33
- The introduction should include more examples of previous similar attempts and make the case why there is a need for new methods of well-being assessment and what this methods should add to previous methods.
The group concurs with the reviewer, as the introduction lacked critical information on prior examples of similar indices. In addition to this comment, other reviewers suggested similar changes. Thus, the introduction was also updated in its entirety in order to incorporate these changes and updated literature. Page 1-3, Lines 37-125
- The methods section is very lengthy and hard to follow. Also there are some components which should be presented in results section not in methods section. Like the number of excluded patients from the study because of different reasons. You have repeated this information in the results section
The commentary by the reviewer is much appreciated and has been considered during the editing of our manuscript. As such, multiple changes to the methods were incorporated throughout the section. Moreover, repeated information between the methods and results section were eliminated and left in their corresponding site as suggested by the reviewer. Pages 3-6 Lines 133-273
- More than a third of your sample has been excluded mostly because of lack of specific data. This is serious and shows low quality in data gathering which can be a source of bios. This should be mentioned as a shortcoming of the present study.
Indeed, the reviewer points to a significant limitation of our study. A comment of this matter has been included in the strengths and limitations section of the manuscript. However, we would like to stress on the quality of data in the SUN project. These are Spanish university graduates which are a reliable source of recurrent information, which has been further supported by previous validations studies. The suggested limitations have been discussed in this manuscript, but we assure the reviewers and editor that the project and our conclusions have sufficient quality and supported by the evidence. Page 13 Line 476-487
- You have used a special sample of only university graduates. This make it hard to extend your results to the whole population. This is a serious sampling limitation of your study which should be mentioned as a shortcoming.
We appreciate the reviewers comments and agree that this is a special sample of participants. However, we would like to stress that this sample restriction was an intended characteristic of the cohort since its inception. Sample restriction is another method of controlling for confounders as described in the book Modern Epidemiology Ver. 4 by Lash T. and collaborators. Furthermore, these methods have been modelled after the Nurses’ Health Study and the Health Professionals Follow-Up Study from the University of Harvard, whom apply similar sample restrictions. Indeed, we cannot and do not intend to present our data as representative of the origin population. However, the estimated associations and their underlying mechanisms remain present in our cohort. Moreover, if these associations were to be true, the estimates within a representative sample would in fact be of greater magnitude as the sample is relatively young, has adequate lifestyles and predominantly reports high HRQoL scores. As such, the impact of lifestyles on a sample of greater variability in terms of SF-36 score, would produce estimations of greater magnitude. We have included a section discussing these limitations in its corresponding section and hope this explanation resolves any doubts as to the validity of our results. Page 13 Line 479-487
- Many sentences in results section should be moved to methods section. Results section should only give the results not clarifying how each results have been calculated. This clarifications should be moved to methods section.
We appreciate the commentary by the reviewer and as previously discussed, results and methods were placed in their corresponding sections. Throughout the methods and results sections.
- In discussion section tell the reader what is the advantage of your methods related to previous methods. Is it simpler to perform? Is it more sensitive?
We have taken the commentary by the reviewer as an opportunity to discuss the advantages of the LWB-I, and as such we appreciate this suggestion. The advantage of the index lies on the relative ease to identify precluded lifestyles which are directly associated with decreases in health and well-being. Previous indices were of abstract nature. The SF-36, despite its widespread use, does not provide information on the reasons an individual is experiencing decreased social functioning, or mental health. Thus, the LWB-I evaluates information on lifestyles with strong associations with the SF-36 measure making them easy to interpret and intuitive for the general population. Page 13 Lines 465-473
- The English text is very hard to understand and follow in its present form
We appreciate and agree with the reviewer. Thus, we have conducted a thorough revision of the manuscript to improve the overall quality of the English text. Furthermore, we sought for help from a native English speaker, coauthor Dr. Hershey, to improve the overall manuscript. Throughout the manuscript
Reviewer 2 Report
In this study the authors aimed to design an index capable of objectively evaluating health and well-being based on pondered associations between key nutritional, metabolic, and lifestyle features with health-related quality of life..
Although the study has the potentiality of being shared with the scientific community, I believe that the manuscript would benefit from a minor revision with the attempt to better support their experimental setting.
1. The theoretical framework is scarce, they should clearly describe the scientific evidence that supports the hypothesis they have raised.
2. Methods section:
- Experimental procedures should be better defined
- More information should be provided about the participants’ characteristics.
- Anthropometric measurements and physical tests presuppose a protocol. This element is missing from the methodological description, which may imply an impossibility of replicating the study due to a lack of clarity in this regard.
3. I would like to see more of the practical implications. Based on the analyzed variables, how the authors intend to use their findings?
Kind regards
Author Response
Reviewer 2
In this study the authors aimed to design an index capable of objectively evaluating health and well-being based on pondered associations between key nutritional, metabolic, and lifestyle features with health-related quality of life..
Although the study has the potentiality of being shared with the scientific community, I believe that the manuscript would benefit from a minor revision with the attempt to better support their experimental setting.
The theoretical framework is scarce, they should clearly describe the scientific evidence that supports the hypothesis they have raised.
Introduction changes theoretical framework is scarce
We appreciate the reviewer’s comments, which coincide with other reviewers. As such, we have conducted a thorough revision and update of the introduction. The revised introduction provides more information on prior lifestyle indices and current evidence on the matters discussed throughout the manuscript. As a result, these changes have substantially improved the manuscript. Page 1-3, Lines 37-125
- Methods section:
- Experimental procedures should be better defined
- More information should be provided about the participants’ characteristics.
- Anthropometric measurements and physical tests presuppose a protocol. This element is missing from the methodological description, which may imply an impossibility of replicating the study due to a lack of clarity in this regard.
The authors of this manuscript appreciate the commentary of the reviewer. Clarification of how anthropometric measurements and physical activity was measured is now included in the new version of the manuscript. We hope that these changes provide sufficient information as to our methods, and the characteristics of the sample regarding their lifestyle habits. Page 3, Lines 131-142
- I would like to see more of the practical implications. Based on the analyzed variables, how the authors intend to use their findings?
Kind regards
The commentary by the reviewer is highly appreciated and has been incorporated in the revised manuscript under the strengths and limitations section. In short, the LWB-I is an initial attempt at providing a general assessment of health for early identification of chronic diseases based on detrimental lifestyles. Thus, its use is primarily as a screening tool of the general population and potentially for researchers seeking to contrast this index with similar lifestyle indices. Page 13 Lines 465-473
Reviewer 3 Report
In this manuscript, the authors developed objective Nutritional well-being index based on the metabolic and lifestyle items which is able to objectively evaluate health conditions of paticipants. The methods and results were well-described. The authors thorougly discussed the result part and limitation of the current study. I am happy to accept this manuscipt in the present form.
Author Response
Reviewer 3
In this manuscript, the authors developed objective Nutritional well-being index based on the metabolic and lifestyle items which is able to objectively evaluate health conditions of paticipants. The methods and results were well-described. The authors thorougly discussed the result part and limitation of the current study. I am happy to accept this manuscipt in the present form.
We appreciate the reviewer’s comments and are pleased to hear that our work holds up to the journals high standards.
Reviewer 4 Report
Thank you for the opportunity to review this paper. The authors have provided an interesting manuscript on the development of a tool for quick assessment of wellbeing.
I have major concerns about the language of the manuscript. The choice of words is poor and sentence structure fails to convey the authors’ meaning well. Past tense should be used consistently when describing completed studies. I would recommend that the authors use professional editing services to ensure that their manuscript is suitable for an English-speaking readership. At present, the manuscript lacks clarity in multiple places, which makes full assessment difficult.
The introduction provides a good explanation of the reasons for developing the tool. However, the last paragraph is confusion as it includes details of the tool that have not been explained. I would suggest that instead of referring to cut-off points and states that are not clear to the reader, the authors explain the aim only at this stage.
The methods include a good variety of relevant lifestyle behaviour and nutrition measures. It is not clear whether the measures were compared for the same time point (e.g. 4 years after graduation) or at the same time (e.g. 2005) for all participants. Please specify whether you used the licensed (Optum) or free (Rand) version of SF-36 for clarity. The use of terminology is also key here – SF-36 covers 8 domains rather than aspects or spheres. A clearer description of the timing of all assessments in one place, preferably as a flowchart, would be helpful for better understanding of the study design and data collection.
The new brief instrument could be useful in screening patient/client wellbeing. Nevertheless, it is not entirely clear why the authors chose to name it ‘nutritional “wellbeing index. The instrument contains a broader range of factors than nutrition and nutritional wellbeing includes factors not included in this instrument, such as food security, economic factors, emotional eating etc. The authors could consider whether something like ‘healthy lifestyle index’ might be more appropriate?
The informed consent statement needs to be revised. At present it sounds like the university committee approved the Declaration of Helsinki.
Author Response
Reviewer 4
Thank you for the opportunity to review this paper. The authors have provided an interesting manuscript on the development of a tool for quick assessment of wellbeing.
I have major concerns about the language of the manuscript. The choice of words is poor and sentence structure fails to convey the authors’ meaning well. Past tense should be used consistently when describing completed studies. I would recommend that the authors use professional editing services to ensure that their manuscript is suitable for an English-speaking readership. At present, the manuscript lacks clarity in multiple places, which makes full assessment difficult.
The group concurs with the reviewer and appreciates the commentary on the quality of the English text of the manuscript. As such, we have conducted a full revision of the manuscript, which was led by a native English speaker and co-author of this work, Dr. Hershey. Throughout the manuscript
The introduction provides a good explanation of the reasons for developing the tool. However, the last paragraph is confusion as it includes details of the tool that have not been explained. I would suggest that instead of referring to cut-off points and states that are not clear to the reader, the authors explain the aim only at this stage.
Introduction: Touch on relevant concepts and points within the LWB-I (Reasons for the use of lifestyles) , inclusion of SPH and the proposed definition.
The commentary by the reviewer is much appreciated and noted by other reviewers as well. Substantial changes have been made to the introduction of the manuscript providing 1) more information on the problem at hand and implications for public health, 2) the advantages of lifestyle assessments and the outcomes of SPH and HRQoL and 3) previous experiences with similar indices. We hope these changes provide sufficient background to the reviewers and readers as to the advantages that the LWB-I provide in the field of research. Page 1-3, Lines 37-125
The methods include a good variety of relevant lifestyle behaviour and nutrition measures. It is not clear whether the measures were compared for the same time point (e.g. 4 years after graduation) or at the same time (e.g. 2005) for all participants. Please specify whether you used the licensed (Optum) or free (Rand) version of SF-36 for clarity. The use of terminology is also key here – SF-36 covers 8 domains rather than aspects or spheres. A clearer description of the timing of all assessments in one place, preferably as a flowchart, would be helpful for better understanding of the study design and data collection.
The concern by the reviewer is acknowledged and appreciated. Our revised manuscript has incorporated a diagram of the time points at which data was collected and analyzed. For clarification, data was analyzed cross-sectionally for participants at their 4th year of follow-up. This is the 4th year of their permanence within the cohort. Given the SUN projects characteristics (a dynamic cohort), this time point varies for all participants. Figure 1 and Page 3 Lines 135-146
The new brief instrument could be useful in screening patient/client wellbeing. Nevertheless, it is not entirely clear why the authors chose to name it ‘nutritional “wellbeing index. The instrument contains a broader range of factors than nutrition and nutritional wellbeing includes factors not included in this instrument, such as food security, economic factors, emotional eating etc. The authors could consider whether something like ‘healthy lifestyle index’ might be more appropriate?
We appreciate the reviewer’s commentary and agree that both patients and clients can be screened with our tool. Moreover, we also agree that the index more closely resembles a lifestyle index. As such, we have decided as a group to change the title of the manuscript to the following: “Development of a general health score based on 12 objective metabolic and lifestyle items: the Lifestyle and Well-Being Index”. We hope these changes appeal to the reviewers and the readers of this notorious journal. Title change
The informed consent statement needs to be revised. At present it sounds like the university committee approved the Declaration of Helsinki.
We apologize for the lack of clarity in the informed consent statement. As such we have changed it to better reflect what the Ethics committee has approved. Ethics statement Page 14
Round 2
Reviewer 1 Report
The manuscript has improved. Still needs some revisions. Language flow is improved but needs more work.
Author Response
Please see the attachement
